# Prevalence of Postural Changes and Musculoskeletal Disorders in Young Adults

**DOI:** 10.3390/ijerph20247191

**Published:** 2023-12-17

**Authors:** Maria Paula Pacheco, Paulo José Carvalho, Luís Cavalheiro, Filipa Manuel Sousa

**Affiliations:** 1Polytechnic Institute of Coimbra, Coimbra Health School, 3046-854 Coimbra, Portugal; luisc@estesc.ipc.pt; 2Polytechnic Institute of Porto, School of Health, 4200-072 Porto, Portugal; paulocarvalho@ess.ipp.pt; 3Biomechanics Laboratory, Faculty of Sport, University of Porto, 4200-450 Porto, Portugal; filipas@fade.up.pt

**Keywords:** postural changes, young adults, musculoskeletal disorders, physiotherapy

## Abstract

Background: Postural changes are considered a public health problem, especially those that affect the spine, as they may predispose to degenerative conditions of the spine in adulthood. Musculoskeletal disorders are the main cause of chronic pain, illness, reduced educational performance, productivity, and quality of life, and are responsible for increased absenteeism, which could compromise the future career of students. The purpose of this study was to identify the prevalence of postural changes and the occurrence of musculoskeletal disorders in different anatomical regions in the 12 months prior and 7 days prior to the application of the questionnaire and the number of affected anatomical regions. Methods: This is an observational, descriptive, cross-sectional study. It included 508 students selected by a stratified random sample. Our outcomes were the Portuguese version of the Standardized Nordic Musculoskeletal Questionnaire, the Adam’s test, a scoliometer, and the visual analog pain scale associated with the Nordic Musculoskeletal Questionnaire. Results: In total, 79.3% of the students tested positive with the Adam’s test. The neck, shoulder, lumbar region, and knee stood out in all of the temporal references, the 12 months prior (44.3%; 35.2%, 50.2%, 34.1%) and the 7 days prior (16.5%, 16.9%; 28.9%, 17.5%), and even in the restriction of activity due to painful symptoms in the 12 months prior (4.3%, 5.3%, 10.6%, 8.5%). Conclusions: Out of 497 students, 403 were identified with postural changes. The high prevalence rate of identified musculoskeletal symptoms in the anatomical regions of the neck, lumbar region, and shoulder raises the need for intervention in students. Gender appears to generate differences between men and women. Pain from multiple body sites is frequent among young adults.

## 1. Introduction

Postural changes can cause increased stress on muscles, ligaments, joints, and bone structures [1]. These are considered a serious public health problem, given their great impact on the population, permanently or temporarily disabling them from carrying out their professional activities [2,3,4,5]. Postural changes largely affect economically active individuals who adopt body postures that are inappropriate for their anatomical structures and who, as a result, end up withdrawing from professional activities, often due to pain, discomfort, or even disability [6]. Most postural problems have their origin during the period of body growth and development, that is, in childhood and adolescence—a period in which anatomical structures undergo an accommodation process. Some of these changes are characteristic of this phase; however, others can negatively impact the quality of life of individuals [7,8,9].

The study of postural changes requires the definition of a reference posture that, in an upright position, concerns the relationship between the line of gravity and the body segments [10]. If a certain region of the body moves forward or behind the line of gravity, all other body regions will compensate to maintain balance, increasing the possibility of acquiring bad posture [11]. For example, the anterior projection of the head can cause an increase in thoracic kyphosis and an anterior positioning of the shoulder [12]. According to the 1947 definition of the Posture Committee of the American Academy of Orthopedics, posture corresponds to the balance of skeletal components to preserve the body’s support structures from injuries and progressive deformations [13,14]. It can also be defined as the position or attitude of the body in a static position or the harmonious combination of different body regions in dynamic situations [5,15].

Good posture is defined as the alignment of the body with maximum physiological and biomechanical efficiencies, which minimizes the stress and overload suffered by the support system due to the effects of gravity [1,16]. A balanced posture protects body structures against injuries or deformities [15]. Incorrect posture can generate muscular imbalances in the affected region, as well as biomechanical compensations in the segments, to keep the gravitational center balanced [5,17]. Incorrect posture has many negative effects on the spine. For example, joint imbalance limits the movement of the tendons and muscles and makes normal exercise and movement difficult. Additionally, incorrect posture can cause pain [18]. Postural changes are considered a public health problem, especially those that affect the spine, as they may predispose to degenerative conditions of the spine in adulthood [19]. Proper posture should be characterized by symmetry in the frontal and transverse planes, and the spinous process line should overlap with the mechanical axis of the spine [20].

Some postural deviations can reduce muscular efficiency, predisposing individuals to pain and pathological musculoskeletal conditions, causing unsightly changes [21,22,23,24]. For example, forward head posture seems to be able to be associated with neck pain and chronic migraine [25,26].

Some studies have reported a high incidence of postural problems. Changes were found in university students in which 97% presented scapula-pelvic asymmetry, 85.7% cervical hyperlordosis, 74.2% forward torso, 65.7% lumbar hyperlordosis, and 100% showed a tendency towards scoliosis [27]. Others identified a prevalence of 57.4% for cervical hyperkyphosis, 83.3% forward head position, 68.5% lumbar hyperlordosis, and 66.6% pelvis anteversion [28]. Another study evaluated head protrusion; cervical lordosis; thoracic kyphosis; lumbar lordosis; pelvic tilt, knee flexion, and tibiotarsal angle in nursing students before and after clinical practice, and the results showed that students in general presented considerable postural changes when compared to the normal standard. The authors concluded that all participants experienced significant postural changes [29].

Musculoskeletal disorders (MSD) are described as permanent injuries or pain in the body that affect muscles, ligaments, joints, bones, nerves, and spinal discs. The most common symptoms of MSD are pain, joint stiffness, tingling, and numbness in muscles, as well as reduced mobility and functional impairment [30,31]. MSDs are significant, extremely common health problems that affect the vast majority of individuals, regardless of age, sex, and sociodemographic level [32]. It is also mentioned that adolescents with complaints of musculoskeletal pain are more likely to develop chronic musculoskeletal pain in adulthood [33]. Analysis of data relating to the global burden of disease revealed that the worldwide prevalence of MSD is 21.9%, affecting all ages with a continuous increase, contributing to 17% of all years lived with disability across the world [34]. Taking Canada as an example, it has a high prevalence of MSD with point prevalence estimates in 2017 of up to 27.8%, with the highest prevalence in the spine and knee joints [35].

Some authors report that MSDs are a major cause of severe pain and disability in the long term, with a loss of productivity and decreased quality of life, which can lead to a reduction in academic performance in students [36,37,38,39,40]. In recent years, MSDs have emerged as a public health problem among university students, where their prevalence has shown alarming numbers, with reference to prevalence rates between 32.9% and 89.3% in several countries [41]. In the literature, it is suggested that being a university student may be the only risk factor for developing MSD, due to the increased time spent in front of a screen and prolonged time in a sitting position, combined with high levels of stress [42]. It is also said that, on the one hand, musculoskeletal disorders are the main cause of chronic pain, illness, reduced educational performance, productivity, and quality of life, and are responsible for increased absenteeism, which could compromise the future career of students. It also negatively affects mood and can trigger irritability, anxiety, depression, disability, and social problems, reducing general health status [43]. Thus, MSDs not only compromise quality of life but also have a negative economic impact on society at the individual level [44].

The importance attributed to the topic is mentioned in the Bull World Health Organ 2018, highlighting that musculoskeletal conditions represent the largest proportion of persistent pain across geographies and ages, and reinforcing that musculoskeletal health is critical for human function, as it allows mobility, dexterity, and ability to work and actively participate in all aspects of life [45,46]. The profile of non-communicable diseases, namely neonatal, maternal, nutritional, and musculoskeletal conditions is changing, as global disability-adjusted life years (DALYs) have increased from 43.9% in 1990 to 61.4% in 2016. This concern regarding the health profile is reflected in the global population with the increase in chronic disease and injuries, mainly musculoskeletal conditions [47,48]. The Sustainable Development Goals (SDGs) and the 2020–2030 Decade for Healthy Aging provide a timely vision and a favorable opportunity to increase the focus on action on musculoskeletal health [49].

The objective of the present study is to identify postural changes and the occurrence of musculoskeletal disorders in different anatomical regions, in the 12 months prior and the 7 days prior to the application of the questionnaire, and the number of affected anatomical regions.

## 2. Materials and Methods

### 2.1. Design Study

This is an observational, descriptive, cross-sectional study carried out at the Health School of Coimbra, where courses in Audiology, Biomedical Laboratory Sciences, Dietetics and Nutrition, Pharmacy, Clinical Physiology, Physiotherapy, Medical Imaging and Radiotherapy, and Environmental Health are taught.

### 2.2. Study Population

The target population was made up of students from the 8 degrees taught at the Coimbra Health School, Polytechnic University of Coimbra, enrolled in the 1st and 2nd years, and aged 18 years old or over. Participants were considered eligible when they agreed to participate in the study and signed the free and informed consent form, in which they showed interest in taking part in this same study. Exclusion criteria were the presence of any pathology or known injury at the level of the musculoskeletal, neuromuscular, cardiorespiratory, or other systems.

### 2.3. Sampling Type and Technique and Sample Size

The study sample is of the probabilistic type and the sampling technique used is stratified random. The sample size and the associated sampling error were calculated considering the following formula:$$e=\sqrt{\frac{\left(\left(N-n\right)\times 1.96^{2}\times 0.5\left(1-0.5\right)\right)}{n\times N}}$$
$$e=\sqrt{\frac{\left(\left(716-508\right)\times 1.96^{2}\times 0.5\left(1-0.5\right)\right)}{508\times 716}}$$

*e* = sampling error; N = target population; *n* = sample; K = 1.96 (standardized normal variable associated with the level of confidence); *p* = 0.5 (probability of the event).

The total sample, for a sampling error of 2.3%, consisted of 508 students, who agreed to participate voluntarily. The sample was stratified by course, year, and gender.

### 2.4. Measuring Instruments

As measuring instruments, the Portuguese version of the Standardized Nordic Musculoskeletal Questionnaire (SNMQ) [50], the Adam’s test, and a scoliometer from the brand GIMA S.p.A-20060 Gessate (MI)-Italy [51,52] and the visual analog pain scale (VAS) associated with the Nordic Musculoskeletal Questionnaire [53] were used.

The Standardized Nordic Musculoskeletal Questionnaire was developed with the purpose of standardizing and measuring symptoms of musculoskeletal origin. The authors do not indicate it as a basis for clinical diagnosis, but for the identification of musculoskeletal disorders, which may constitute an important diagnostic tool. It can take three formats: (i) general, comprising 9 anatomical areas (neck, shoulders, elbows, wrist/hands, dorsal and lumbar region, hips/thighs, knees, and ankles/feet); (ii) specific for the lumbar region; (iii) specific to the neck and shoulders [54].

In the present study, only the general format was used, which evaluates, more specifically, the occurrence of musculoskeletal disorders in the nine anatomical areas already mentioned, considering the 12 months prior, the 7 days prior, and the reduction in activity caused by them in the precedent 12 months. It is quoted individually for each region. For the neck, dorsal region, lower back, hips/thighs, knees, and ankles/feet, the answer options are included on a dichotomous scale where 1 = no and 2 = yes. In areas of the upper limb, lateral or bilateral discrimination is allowed: 1 = no; 2 = yes on the right; 3 = yes on the left; and 4 = yes, both [54].

This instrument was adapted and validated for the Portuguese culture, with a Cronbach’s Alpha value of 0.924 for internal coherence and r values ranging between 0.677 and 1 for inter-temporal reliability. Criterion validity was defended based on the relationships observed with the Oswestry Disability Index (r between 0.290 and 0.479; *p* < 0.05 and 0.001) [50].

The Adams Test was used to measure spinal deformity, performed through forward flexion of the torso. This test has a sensitivity of 92% and a specificity of 60% in diagnosing thoracic scoliosis [52].

The scoliometer allows for the identification of the scoliotic curvature, avoiding the exposure of patients with scoliosis to radiation, ease of evaluation in the office, and a reduced cost, compared to radiological examination. It is a device that contains a metal sphere inside, soaked in water, which indicates the torso’s axial rotation angle, and that can be moved in a range of 0° to 30° to both sides in an increasing scale of unitary values. It is a reliable tool to assess axial torso rotations in individuals with idiopathic scoliosis in all segments of the spine, especially if the assessment is performed by the same evaluator and in the middle and lower thoracic segments. The evaluator positions the scoliometer perpendicularly to the axial axis of the spine on the spinous processes of the vertebrae, leveled with the marking referring to the center of the scoliometer [51]. In a study by Côté et al. [52], inter-rater reliability values of 0.91 were obtained for the thoracic region and 0.74 for the lumbar region, with the examiners reproducing the entire assessment, from the individual’s positioning, determination of vertebrae, and recording the measurement with the scoliometer. This instrument is described as a highly reliable instrument in both inter- and intra-rater analyses (r = 0.86–0.97) [55].

The Visual Analog Scale is a one-dimensional pain intensity assessment instrument that is represented by a straight horizontal unnumbered line measuring 10 cm in length. This line consists of two ends numbered 0 and 10 (where 0 corresponds to a total absence of pain and 10 to the worst imaginable pain). The individual will be asked “How bad is your pain?” and after the question, they will be instructed to mark the same with a vertical line at the point that represents the intensity of their pain felt at the moment. Subsequently, the numerical value marked by the individual will be measured using a ruler. Regarding its psychometric properties, this scale indicates good levels of reliability, as it presents a value of r = 0.94 in the test–retest and a construction validity that varies between r = 0.62 and r = 0.91 [53].

### 2.5. Information Collection Methods

After obtaining authorization to carry out the study from the presidency of Coimbra health school, the academic services were asked to provide a list of students enrolled in the 1st and 2nd years of the aforementioned degrees, through which it was possible, with the collaboration of the department directors, to have access to the timetables and contact teachers to proceed, before or after classes, to explain the objectives of the study, inclusion and exclusion criteria, time required to complete a questionnaire that included the collection of sociodemographic and anthropometric data, intensity of pain felt at the time, presence or absence of musculoskeletal problems, existence or non-existence of restriction of activities due to these problems, and carrying out a clinical assessment to identify the presence of postural changes ascertained with a simple anterior torso tilt test (Adams test) after which, if it were positive, a measurement with a scoliometer would be applied to evaluate and quantify humps (Figure 1).

### 2.6. Statistical Analysis

Statistical analysis of the data was performed using the statistical software IBM SPSS (Statistical Package for Social Sciences), version 28.0.1.0 for Windows.

Descriptive statistics were used for the characterization and general description of the sample, through frequencies and respective percentages and statistical measures of central tendency (mean) and dispersion (amplitude and standard deviation). Comparison between groups (gender) was evaluated using the chi-square test, given the normality of the sample. Statistical significance for *p* values < 0.005.

### 2.7. Ethical Issues

Potential candidates were informed of the voluntary nature of their participation and if they agreed to participate, they would be asked to sign the free and informed consent form. Participants were also informed about the objectives and conditions of the study and the possibility of abandoning it at any time, if they so chose. The principles of Helsinki declaration were followed and the study was approved by the Ethics Committee of the Faculty of Sports of the University of Porto (CEFADE 17.2019) on 17 July 2019.

## 3. Results

### Characterization and General Description of the Sample

This study had the participation of 508 individuals, mostly female (78.9%). Of these, 497 were evaluated with the Adams test, with 403 individuals testing positive (79.3%), and 39% presenting a hump greater than or equal to 5 degrees.

Students had an average age of 19.41 ± 1.59 years, a body mass index varying between a minimum of 13.27 and a maximum of 38.57, and an average intensity of pain of 3.57 ± 2.80 (Table 1).

Regarding the presence or absence of musculoskeletal problems in the prior 12 months and prior 7 days, as well as the existence of activity restrictions in the last year, Table 2 reports pain, discomfort, or numbness in the prior 12 months and prior 7 days in nine regions of the human body, as well as the existence of activity restrictions in the last year. From this observation, it is possible to infer that the highest values of the prevalence of this symptomatology concern the neck (44.3%) and the lumbar region (50.2%), which is also the region indicated with a higher percentage in terms of activity restriction in the last year. The prevalence rates for the shoulder and knee presented percentages of 35.2% and 34.1%, respectively. When we look at the same symptomatology, but now considering the last 7 days, we find that the lumbar region continues to be the most marked (29.8%), followed by the knee, shoulder, and finally the neck (Table 2).

When we analyze the data by sex, there appears to be statistically significant differences between men and women in most of the anatomical regions observed in the 12 months prior to the application of the questionnaire. In the 7 days prior to the application of the questionnaire, differences were only observed for the thoracic and lumbar regions (*p* = 0.004 and *p* < 0.001, respectively). There are no differences between sexes in activity restrictions (Table 2).

In the prior 12 months, students reported an average of 2.5 ± 2.02 anatomical regions with pain, which dropped to 1.21 ± 1.48 in the last 7 days and presented a residual average in terms of activity restriction (0.43 ± 0.82) (Figure 2).

## 4. Discussion

This study aimed to identify postural alterations, the occurrence of musculoskeletal disorders in nine anatomical regions in the prior 12 months and prior 7 days, the existence of restriction of activities in the last year, and even the number of anatomical regions highlighted in Coimbra Health School students attending the first and second years of the degrees taught there. Briefly, we can say that our sample consisted of 508 higher education students, mostly female and with an average age of 19.41 ± 1.59 years.

This profile is generally included in that described for university students. These data are in line with data from “PORDATA statistics in Portugal and Europe” (https://www.pordata.pt/en/europe; accessed on 24 July 2023), which states that in 2022, 233,747 women and 199,470 men enrolled in higher education. Regarding the average age, we can say that this is on average, normalized, since the most frequent scenario is for students to start the course at 17/18 years old, so in the first 2 years of the course the average age obtained is natural. In order to obtain a representative sample of the population, the calculation was carried out for a sample with an associated error that was acceptable (2.3%), to guarantee this representativeness. The sample was also stratified to ensure relative representativeness, depending on the degree course, the year of the respective course, and the gender of the students. However, during the collection of the sample elements, we had to choose to guarantee the entire sample instead of the stratification initially carried out, since it was not possible in some situations to obtain enough elements to fully complete the values foreseen in the stratification, being replaced by others, until obtaining the total value of the sample. In any case, the final values obtained, either by course or by sex, do not differ substantially in percentage terms, which is why the values obtained were considered to be acceptable, from the point of view of relative representativeness.

In our study, we performed the Adams test to assess the existence of postural changes, as it is a quick and reliable test in its detection. We found a high prevalence in which 79.3% of the students tested positive. This result is in line with the prevalence rates described in the literature that vary between 32.9 and 89.3% [41]. Some examples are the studies by Sertarath and Tuza which reveal percentages of 73.6 and 89.5, respectively [56,57]. The explanation for these results of postural changes in university students may be associated with physical inactivity and the sitting position maintained for long periods which, in addition to compensations and postural changes, can cause muscle overload, muscle fatigue, and, consequently, compression of blood vessels and nerve endings, culminating in pain, mainly in the spine [11,27,58]. According with what was previously written, young adulthood could be the crucial time to develop intervention programs in terms of health promotion, particularly in terms of musculoskeletal disorders.

A moderate pain intensity, scored on average at 3.57 points, seems a bit high for the study population. However, in analyzing the data more carefully, the explanation we found can be due to the high variability observed, reflected at a standard deviation of ±2.80, as well as the context in which the question was asked, regarding the worst pain felt at the time of data collection.

In relation to the analysis of the prevalence of MSD in the different anatomical regions (the main objective of our study), we can see that in the students in the sample, the neck, shoulder, lumbar region, and knee stood out in all the temporal references, in the prior 12 months (44.3%; 35.2%, 50.2%, 34.1%), the prior 7 days (16.5%, 16.9%; 28.9%,17.5%), and even in the restriction of activity due to painful symptoms in the 12 months prior (4.3%, 5.3%, 10.6%, 8.5%). These results show a high prevalence of MSD as also described in other studies carried out in several countries among university students in the health field, such as Saudi Arabia [59], Malaysia [60], Iran [61], Brazil [62,63], Croatia [64], Italy [65], South Africa [66,67], Israel [68], Sri Lanka [56], Ethiopia [69], and also the studies by Almhdawi et al. and Hasan et al. [70,71] carried out in students of the health area.

If we look at the different international realities, and regarding symptoms in the last year, several authors also report high prevalence rates for the neck area, which can vary between 34 and 88% [63,68]; for the shoulder between 11 and 63.6% [65,67]; the same can be verified for the lumbar region with values described between 27 and 81.1% [65,67]; and for the knees varying between 25 and 44.1% [68,71]. In our study, symptoms in the lumbar region were the most prevalent ones. This fact did not surprise us, given that this is considered the most prevalent musculoskeletal disorder [72], and given the knowledge that approximately 80% of people experience low back pain at least once during their lifetime [73].

When we compare by gender, the main differences appear in the most of anatomical regions in the last 12 months [65,67]. This suggests that females report more pain or discomfort than males. It is known that gender is a key factor in musculoskeletal disorder and postural alterations, so this result is expectable and in accordance with other studies which showed a higher prevalence of musculoskeletal pain symptoms in women in the neck, shoulders, upper back, and lower back when compared to men [64,74].

In the 12 months prior to the application of the questionnaire, students reported an average of 2.5 simultaneously symptomatic anatomical regions, which dropped to an average of 1.21 regions in the 7 days prior, and presented a residual average in terms of activity restriction. Similar values were found in the study by Netanely and Parto et al. in which recording the number of anatomical regions with pain varied between 2 and 4 [68,75]. It seems important to reflect this result since this pattern of multiple pain locations can persist over time. This is a situation that may have future implications, so planned interventions must be considered, both in terms of prevention and treatment.

The data were collected through self-report questionnaires, which may result in under- or overestimation of MSD and the frequency of MSD complaints was not compared to a control group, which could be identified as a limitation. The study also used a sample exclusively from the Coimbra Health School, and it is not possible to guarantee the generalization of the results. Future studies identifying predictive factors for the occurrence of postural changes, as well as the use for more precise assessment instruments, are recommended. However, attending to the worldwide reality, these results could contribute to the definition of health-promotion and prevention strategies in the field of occupational health and physiotherapy intervention related to musculoskeletal disorders.

## 5. Conclusions

Out of 497 students, 403 were identified with postural changes. The high prevalence rate of identified musculoskeletal symptoms in the anatomical regions of the neck, lumbar region, and shoulder raises the need for intervention in students. Gender appears to generate differences between men and women. Pain from multiple body sites is frequent among young adults.

Thus, it is suggested that screening be carried out to identify risk factors and the use of health-promotion strategies, namely initiatives that promote programs for the prevention of musculoskeletal disorders in students.

## Figures and Tables

**Figure 1 ijerph-20-07191-f001:**
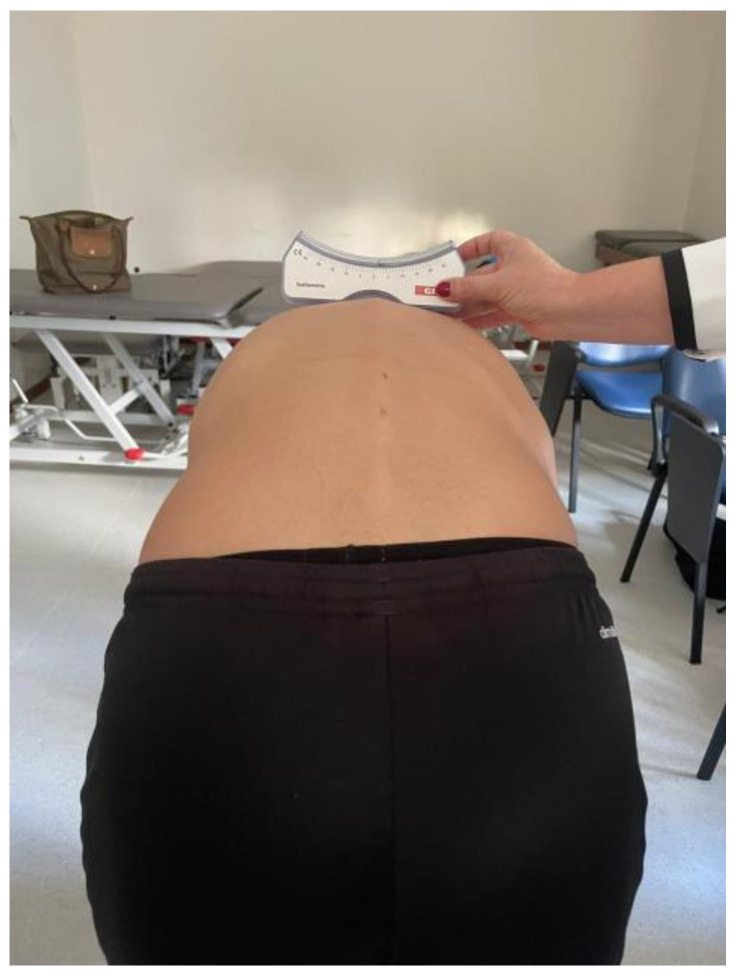
Measurement with a scoliometer.

**Figure 2 ijerph-20-07191-f002:**
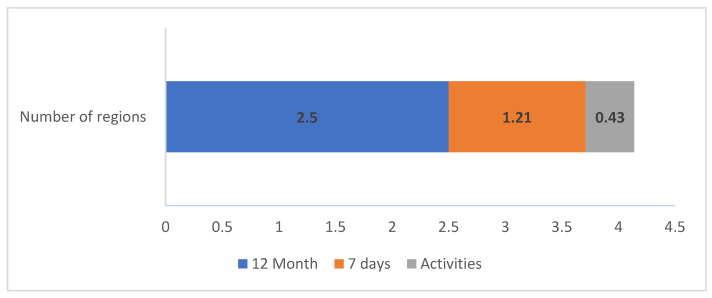
Number of anatomical regions involved.

**Table 1 ijerph-20-07191-t001:** Sociodemographic, anthropometric characteristics, and pain intensity.

	*n*	Min	Max	Mean	SD
Age (years)	506	18	34	19.41	1.59
Weight (kg)	500	37.00	115.00	61.35	11.26
Height (m)	502	1.48	1.92	1.66	0.08
BMI (kg/m^2^)	497	13.27	38.57	22.09	3.11
VAS	503	0.0	10.0	3.57	2.80

**Table 2 ijerph-20-07191-t002:** Positive answers to different dimensions of the Nordic Musculoskeletal Questionnaire. Males vs. females (*n* = 508).

	Total		Female		Male		
Pain, Discomfort, or Numbness Last 12 Months (*n* = 508)	*n*	%	*n*	%	N	%	*p* *
Neck	225	44.3	175	43.6	50	46.7	---
Shoulder	179	35.2	152	37.9	27	25.2	0.015
Elbow	23	4.5	20	5.0	3	2.8	---
Wrist/Hand	129	25.4	102	25.4	27	25.2	0.001
Thoracic Region	101	19.9	88	21.9	13	12.1	0.024
Lumbar Region	255	50.2	217	54.1	38	35.5	<0.001
Hip/Thigh	89	17.5	75	18.7	14	13.1	---
Knee	173	34.1	150	37.4	23	21.5	0.002
Foot	96	18.9	79	19.7	17	15.9	---
Pain. Discomfort, or numbness last 7 days (*n* = 508)							
Neck	84	16.5	68	17.0	16	15.0	---
Shoulder	86	16.9	71	17.7	15	14.0	---
Elbow	14	2.8	13	3.2	1	0.9	---
Wrist/Hand	50	9.8	38	9.5	12	11.2	---
Thoracic Region	56	11.0	50	12.5	6	5.6	0.044
Lumbar Region	147	28.9	134	33.4	13	12.1	<0.001
Hip/Thigh	46	9.1	38	9.5	8	7.5	---
Knee	89	17.5	75	18.7	14	13.1	---
Foot	43	8.5	38	9.5	5	4.7	---
Activity Restriction 12 month (*n* = 508)							
Neck	22	4.3	18	4.5	4	3.7	---
Shoulder	27	5.3	21	5.2	6	5.6	---
Elbow	3	0.6	3	0.7	0	0.	---
Wrist/Hand	19	3.7	15	3.7	4	3.7	---
Thoracic Region	22	4.3	17	4.2	5	4.7	---
Lumbar Region	54	10.6	47	11.7	7	6.5	---
Hip/Thigh	17	3.3	13	3.2	4	3.7	---
Knee	43	8.5	35	8.7	8	7.5	---
Foot	30	5.9	23	5.7	7	6.5	---

* Chi square differences between males and females. --- Not significant.

## Data Availability

The data that support the findings of this study are available on request from the corresponding author. The data are not publicly available due to privacy and ethical restrictions.

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
