# Peer review of "Prevalence of Postural Changes and Musculoskeletal Disorders in Young Adults"

_ijerph, 2023, doi:10.3390/ijerph20247191_

Round 1
Reviewer 1 Report
Comments and Suggestions for Authors
Dear authors,
Thank you for your efforts and diligence in your research. Your research is valuable in terms of its subject and scope, and your article is very suitable for a prevalence study. My only concern for your research is that although it is a prevalence study, the number of subjects is small. However, you have exceeded this situation by calculating the number of subjects.
You should thoroughly revise your research by paying attention to journal writing rules. I recommend that you pay attention to the spelling and where subheadings require numbering.
Also, please support your physical measurements with photographs so that we can understand them better.
After all these minor additions, your research is suitable for publication in IJERPH.
Author Response
Dear authors,
Thank you for your efforts and diligence in your research. Your research is valuable in terms of its subject and scope, and your article is very suitable for a prevalence study. My only concern for your research is that although it is a prevalence study, the number of subjects is small. However, you have exceeded this situation by calculating the number of subjects.
# Authors' Answers
The authors are grateful for the reviewer's comments, which will certainly contribute to improving the final version of the manuscript.
You should thoroughly revise your research by paying attention to journal writing rules. I recommend that you pay attention to the spelling and where subheadings require numbering.
# Authors' Answers
The article was completely revised in terms of spelling corrections and the numbering of subheadings in accordance with the journal writing rules.
Also, please support your physical measurements with photographs so that we can understand them better.
# Authors' Answers
A representative photograph of the scoliometer measurement was introduced.
After all these minor additions, your research is suitable for publication in IJERPH.
Reviewer 2 Report
Comments and Suggestions for Authors
Dear Authors, thank you for the opportunity to read the manuscript. The topic of musculoskeletal disorders among young adults is an important issue, so I read the manuscript with attention and enthusiasm. Unfortunately, it lacks a lot of information, which in my opinion is crucial in this type of publication. Below I explained my concerns in detail:
It is necessary to extend the introduction to include information on the impact of poor posture on the body and health. It has been treated very briefly, it would be worth giving examples to make the reader aware of the need to maintain the correct posture as incorrect posture does not necessarily result in the appearance of MSDs.
It is not clear why modern or severe pain was considered an exclusion criterion. After all, it can also be a symptom of MSDs, and thus the authors have removed these potential participants from the available pool. It is all the more surprising that in Table 1 we have information that the maximum pain reported by participants was 10 on the VAS scale. It seems that, according to the criteria established by the authors, such a participant should not be included in the study.
„Incorrect posture can generate muscular imbalances in the affected region, as well as biomechanical compensations in the segments, to keep the gravitational center balanced” - it is worth noting, that muscular imbalances can generate incorrect posture too.
„Out of 497 students, 401 were identified with postural changes.” - Where did the number of 497 come from, since 508 students took part in the study and this is the number listed in the results tables?
There should be a comparison of women and men, the results between those groups may vary, they may indicate different needs, including the needs for physical activity and health education between these groups.
„these results could contribute to the definition of health promotion and prevention strategies in the field of occupational health and physiotherapy intervention related to musculoskeletal disorders.” - In my opinion, this is a controversial statement. The authors showed only the occurrence of MSDs. On the basis of their results, few practical conclusions can be drawn to enable appropriate action on the part of public health institutions or bodies. Most of the papers cited by the authors in which similar studies were conducted in other countries took into account factors such as physical activity, history of previous injuries, substance abuse, and other factors. In abovementioned studies the relationship between these factors and the occurrence of MSDs has often been examined. However, the authors conduct almost no analysis of the data they collected, giving only the percentage of MSDs in different parts of the body. For example, the effect of BMI or sex on the occurrence of MSD has not been evaluated, although such data have been collected. This issue significantly reduces the practical value of this manuscript, especially since the authors already pointed out in the introduction that studying at a university according to the literature is a risk factor for MSDs. As such, it seems that the publication does not add much new to the already existing knowledge.
Minor (editorial) issues
The authors explain again the previously explained abbreviations in several places, this should be corrected and the abbreviation explained only at the point of first application. There is also an unexplained abbreviation (most likely a typo) – DME.
Table 2 - Activity Restriction 12 meses – I believe it should be 12 months
„This profile is generally included in that described for university students. These data are
in line with data from “PORDATA statistics on Portugal and Europe”, which states that
in 2022 233,747 women and 199,470 men enrolled in higher education.”– there should be citation
Author Response
Dear Authors, thank you for the opportunity to read the manuscript. The topic of musculoskeletal disorders among young adults is an important issue, so I read the manuscript with attention and enthusiasm. Unfortunately, it lacks a lot of information, which in my opinion is crucial in this type of publication. Below I explained my concerns in detail:
# Authors' Answers
The authors are grateful for the reviewer's comments, which will certainly contribute to improving the final version of the manuscript.
It is necessary to extend the introduction to include information on the impact of poor posture on the body and health. It has been treated very briefly, it would be worth giving examples to make the reader aware of the need to maintain the correct posture as incorrect posture does not necessarily result in the appearance of MSDs.
# Authors' Answers
The introduction has been revised and new information included.
It is not clear why modern or severe pain was considered an exclusion criterion. After all, it can also be a symptom of MSDs, and thus the authors have removed these potential participants from the available pool. It is all the more surprising that in Table 1 we have information that the maximum pain reported by participants was 10 on the VAS scale. It seems that, according to the criteria established by the authors, such a participant should not be included in the study.
# Authors' Answers
We agree with the comment, it is a mistake since this is just a first topic of a broader study within the scope of the main author's doctorate. The criterion concerns a more advanced phase of the study. The pain criterion was removed from the exclusion factors.
„Incorrect posture can generate muscular imbalances in the affected region, as well as biomechanical compensations in the segments, to keep the gravitational center balanced” - it is worth noting, that muscular imbalances can generate incorrect posture too.
# Authors' Answers
The sentence has been clarified.
„Out of 497 students, 401 were identified with postural changes.” - Where did the number of 497 come from, since 508 students took part in the study and this is the number listed in the results tables?
# Authors' Answers
We considered that this interpretation could not be understood clearly. The study sample was in fact 508 individuals. However, when carrying out the Adams test to evaluate changes in the spine, 11 individuals were identified as missing data, resulting in a total of 497 individuals for this variable.
The text was reformulated to clarify this.
There should be a comparison of women and men, the results between those groups may vary, they may indicate different needs, including the needs for physical activity and health education between these groups.
# Authors' Answers
Table 2 was changed to present the comparison between men and women.
„these results could contribute to the definition of health promotion and prevention strategies in the field of occupational health and physiotherapy intervention related to musculoskeletal disorders.” - In my opinion, this is a controversial statement. The authors showed only the occurrence of MSDs. On the basis of their results, few practical conclusions can be drawn to enable appropriate action on the part of public health institutions or bodies. Most of the papers cited by the authors in which similar studies were conducted in other countries took into account factors such as physical activity, history of previous injuries, substance abuse, and other factors. In abovementioned studies the relationship between these factors and the occurrence of MSDs has often been examined. However, the authors conduct almost no analysis of the data they collected, giving only the percentage of MSDs in different parts of the body. For example, the effect of BMI or sex on the occurrence of MSD has not been evaluated, although such data have been collected. This issue significantly reduces the practical value of this manuscript, especially since the authors already pointed out in the introduction that studying at a university according to the literature is a risk factor for MSDs. As such, it seems that the publication does not add much new to the already existing knowledge.
# Authors' Answers
In fact, this is just a small and first part of a broader study, carried out as part of a doctorate, thus including different phases:
First, we carried out a study to evaluate the prevalence of musculoskeletal injuries and postural changes in 1st and 2nd year undergraduate students at Coimbra health school (present paper). From here we moved towards identifying predictive factors for these changes.
Then we have studied the changes using movement analysis instruments, for a more accurate identification of postural changes.
We also carried out an intervention to address these changes through a study with a control group.
Minor (editorial) issues
The authors explain again the previously explained abbreviations in several places, this should be corrected and the abbreviation explained only at the point of first application. There is also an unexplained abbreviation (most likely a typo) – DME.
# Authors' Answers
The entry of abbreviations in the text and their duplications have been corrected.
The abbreviation DME, is a drafting error, should be MSD. It has been corrected in the text.
Table 2 - Activity Restriction 12 meses – I believe it should be 12 months
# Authors' Answers
The correction has been made.
„This profile is generally included in that described for university students. These data are
in line with data from “PORDATA statistics on Portugal and Europe”, which states that
in 2022 233,747 women and 199,470 men enrolled in higher education.”– there should be citation
# Authors' Answers
the statement has been referenced.
Reviewer 3 Report
Comments and Suggestions for Authors
The article presents a topic of indisputable relevance, which is related to WHO priorities and current trends determined by modern life habits and reduced physical activity. This is an observational, cross-sectional study, the main purpose of which was to determine postural changes and disorders of the musculoskeletal system in different anatomical regions, in order to assess the intensity of pain and the number of affected anatomical areas. The authors perfectly put together the most relevant highlights of the topic in the introduction section: these include biomechanical and anatomical aspects, physical and social development, and the quality of life and the role of self-esteem. Study population- young adults, i.e. students aged 18 years or over. The instruments used in the research are considered "soft" and subjective, therefore the results obtained are usually treated as primary, which should be paid attention to by specialists related to this sensitive area, i.e. for educators, health care professionals, physiotherapists and the like.
I get the impression that the article is not written coherently. The first part of it is very strong, but later in the methodology, there is a real lack of justification and confirmation of preferences. Therefore, I will write my comments below in question form:
1) Why did you choose the specified evaluation tools - questionnaires, and no other tools, for example, more suitable for quantitative, more data-confirming analysis? I will try to clarify my question. For example, if you measure something subjectively with a questionnaire, then maybe something can be evaluated quantitatively and more objectively and confirm or link your results in this way. the choice of methods and tools is basically unclear to me. Did you do it because others are doing it? So, what's new from the aggregated data?
2) It seems to me that the statistical analysis is not complete. That's the feeling. You calculated the sample size, but more like nothing much was done, only averages, SD and percentages. How was the data you collected distributed?
3) Looking at the results presented in the tables, there are also large scatters in the data. Perhaps if you analyzed your results more, for example by linking/correlating them to each other, for example, the domains of anthropometry and pain, you could also indicate their connections and give more meaning.
4) I think that the publication needs a graphic implementation: for example, it is possible to show changes in the number of anatomical areas over time or changes in the pain scale in sensitive areas. This way is much clearer and clearer for the reader.
5) Conclusions and abstract state “Out of 497 students, 401 were identified with postural changes”, why not from 508 students?
6) I think more areas of practical value need to be pointed out. What else is this work valuable to a scientific and practical audience?
Minor note: 6 p, in table 2 you left “meses”, correct into months.
Author Response
The article presents a topic of indisputable relevance, which is related to WHO priorities and current trends determined by modern life habits and reduced physical activity. This is an observational, cross-sectional study, the main purpose of which was to determine postural changes and disorders of the musculoskeletal system in different anatomical regions, in order to assess the intensity of pain and the number of affected anatomical areas. The authors perfectly put together the most relevant highlights of the topic in the introduction section: these include biomechanical and anatomical aspects, physical and social development, and the quality of life and the role of self-esteem. Study population- young adults, i.e. students aged 18 years or over. The instruments used in the research are considered "soft" and subjective, therefore the results obtained are usually treated as primary, which should be paid attention to by specialists related to this sensitive area, i.e. for educators, health care professionals, physiotherapists and the like.
# Authors' Answers
The authors are grateful for the reviewer's comments, which will certainly contribute to improving the final version of the manuscript.
I get the impression that the article is not written coherently. The first part of it is very strong, but later in the methodology, there is a real lack of justification and confirmation of preferences. Therefore, I will write my comments below in question form:
1) Why did you choose the specified evaluation tools - questionnaires, and no other tools, for example, more suitable for quantitative, more data-confirming analysis? I will try to clarify my question. For example, if you measure something subjectively with a questionnaire, then maybe something can be evaluated quantitatively and more objectively and confirm or link your results in this way. the choice of methods and tools is basically unclear to me. Did you do it because others are doing it? So, what's new from the aggregated data?
# Authors' Answers
In fact, this is just a small and first part of a broader study, carried out as part of a doctorate, thus including different phases:
First, we carried out a study to evaluate the prevalence of musculoskeletal injuries and postural changes in 1st and 2nd year undergraduate students at Coimbra health school (present paper). From here we moved towards identifying predictive factors for these changes.
Then we have studied the changes using movement analysis instruments, for a more accurate identification of postural changes.
We also carried out an intervention to address these changes through a study with a control group.
2) It seems to me that the statistical analysis is not complete. That's the feeling. You calculated the sample size, but more like nothing much was done, only averages, SD and percentages. How was the data you collected distributed?
# Authors' Answers
Considering the scope of this part of the study, which we explained in the previous answer, the data was treated in accordance with the objectives of the study. Nevertheless, we introduced a comparison between men and women (table 2).
3) Looking at the results presented in the tables, there are also large scatters in the data. Perhaps if you analyzed your results more, for example by linking/correlating them to each other, for example, the domains of anthropometry and pain, you could also indicate their connections and give more meaning.
# Authors' Answers
Considering the explanation given in question 1), this issue will be developed in subsequent studies.
4) I think that the publication needs a graphic implementation: for example, it is possible to show changes in the number of anatomical areas over time or changes in the pain scale in sensitive areas. This way is much clearer and clearer for the reader.
# Authors' Answers
We changed table 3 to a graph, in order to make the information more understandable.
5) Conclusions and abstract state “Out of 497 students, 401 were identified with postural changes”, why not from 508 students?
# Authors' Answers
We considered that this interpretation could not be understood clearly. The study sample was in fact 508 individuals. However, when carrying out the Adams test to evaluate changes in the spine, 11 individuals were identified as missing data, resulting in a total of 497 individuals for this variable.
The text was reformulated to clarify this.
6) I think more areas of practical value need to be pointed out. What else is this work valuable to a scientific and practical audience?
# Authors' Answers
Although we understand this work as part of a larger study, in our opinion, its specific, scientific and practical value has to do with identifying the prevalence of Musculoskeletal injuries and postural changes, in a very precise context, young students ofhealth area in Portuguese higher education.
Minor note: 6 p, in table 2 you left “meses”, correct into months.
# Authors' Answers
The correction has been made.
Reviewer 4 Report
Comments and Suggestions for Authors
The study reports postural alterations in students that is an important issue, nevertheless some points should be reviewed.
Introduction
Please add more references to stress the relationship between postural alteration and pain disorder, for example forward head posture seem to be associated with primary headache, migraine and tension type, see the follow article
- Deodato M, Granato A, Borgino C, Galmonte A, Manganotti P. Instrumental assessment of physiotherapy and onabolulinumtoxin-A on cervical and headache parameters in chronic migraine. Neurol Sci. 2022 Mar;43(3):2021-2029. doi: 10.1007/s10072-021-05491-w. Epub 2021 Aug 5. PMID: 34355296; PMCID: PMC8860953.
- Deodato M, Guolo F, Monticco A, Fornari M, Manganotti P, Granato A. Osteopathic Manipulative Therapy in Patients With Chronic Tension-Type Headache: A Pilot Study. J Am Osteopath Assoc. 2019 Aug 12. doi: 10.7556/jaoa.2019.093. Epub ahead of print. PMID: 31404469.
Add other article that evaluate objective postural alterations
The introdution is to long and confuse please revise all section and follow this line: topic, what we know, gap, aim
Method
Gender play a pivot role in muscleskeletal disorder and postural alterations, I think that is important to add a stratification for gender
why did not use an objective evaluation, such as a phogrammetric postural evaluation?
Result
Why did not report all outcome mesures used? What about scoliometer?
Please add some figure
Discution
These part must be improved, becouse it is difficult to read.
Please revise all discution and follow this line: main findings, comparing your results with others studies, suggesting explanation, limitation, implications
The litation section is to small, the gender is the main limit, the use of only questionare is an other important limitation
Comments on the Quality of English LanguageThe papare needs minor editing english
Author Response
The study reports postural alterations in students that is an important issue, nevertheless some points should be reviewed.
# Authors' Answers
The authors are grateful for the reviewer's comments, which will certainly contribute to improving the final version of the manuscript.
Introduction
Please add more references to stress the relationship between postural alteration and pain disorder, for example forward head posture seem to be associated with primary headache, migraine and tension type, see the follow article
- Deodato M, Granato A, Borgino C, Galmonte A, Manganotti P. Instrumental assessment of physiotherapy and onabolulinumtoxin-A on cervical and headache parameters in chronic migraine. Neurol Sci. 2022 Mar;43(3):2021-2029. doi: 10.1007/s10072-021-05491-w. Epub 2021 Aug 5. PMID: 34355296; PMCID: PMC8860953.
- Deodato M, Guolo F, Monticco A, Fornari M, Manganotti P, Granato A. Osteopathic Manipulative Therapy in Patients With Chronic Tension-Type Headache: A Pilot Study. J Am Osteopath Assoc. 2019 Aug 12. doi: 10.7556/jaoa.2019.093. Epub ahead of print. PMID: 31404469.
Add other article that evaluate objective postural alterations
The introdution is to long and confuse please revise all section and follow this line: topic, what we know, gap, aim
# Authors' Answers
The introduction has been revised with the aim of making it more coherent and clearer. New bibliographical references were also revised and added.
Method
Gender play a pivot role in muscleskeletal disorder and postural alterations, I think that is important to add a stratification for gender
why did not use an objective evaluation, such as a phogrammetric postural evaluation?
# Authors' Answers
we introduced a comparison between men and women (table 2).
In fact, this is just a small and first part of a broader study, carried out as part of a doctorate, thus including different phases:
First, we carried out a study to evaluate the prevalence of musculoskeletal injuries and postural changes in 1st and 2nd year undergraduate students at Coimbra health school (present paper). From here we moved towards identifying predictive factors for these changes.
Then we have studied the changes using movement analysis instruments, for a more accurate identification of postural changes.
We also carried out an intervention to address these changes through a study with a control group.
Result
Why did not report all outcome mesures used? What about scoliometer?
Please add some figure
# Authors' Answers
This is an error on our part, the data for measuring humpness was added to the text. A photo of the measurement with the scolimeter was added in the methods section.
Discution
These part must be improved, becouse it is difficult to read.
Please revise all discution and follow this line: main findings, comparing your results with others studies, suggesting explanation, limitation, implications
# Authors' Answers
The discussion has been revised with the aim of making it more coherent and clearer, also we have introducing the analysis of the comparison between gender.
The litation section is to small, the gender is the main limit, the use of only questionare is an other important limitation
# Authors' Answers
We have improved the limitations of study.
Comments on the Quality of English Language
The papare needs minor editing English
# Authors' Answers
We have done a general correction of English.
Round 2
Reviewer 2 Report
Comments and Suggestions for Authors
The authors addressed all of my comments and made changes to the paper; however, my main concern regarding the lack of risk factor analysis has not been fully addressed. Consequently, the practical value of the work appears to be limited. The authors lack data on other risk factors but conducted an analysis based on sex, while still omitting the second potential factor they investigated, which is BMI. I suggest conducting a similar analysis as done for sex, taking into account BMI (e.g., categorizing patients into overweight, normal, and underweight groups).
Author Response
Comments and Suggestions for Authors
The authors addressed all of my comments and made changes to the paper; however, my main concern regarding the lack of risk factor analysis has not been fully addressed. Consequently, the practical value of the work appears to be limited. The authors lack data on other risk factors but conducted an analysis based on sex, while still omitting the second potential factor they investigated, which is BMI. I suggest conducting a similar analysis as done for sex, taking into account BMI (e.g., categorizing patients into overweight, normal, and underweight groups).
# Authors' Answers
The authors would like to thank the reviewer again for his comments.
In relation to the question raised, we carried out the analysis by BMI, categorizing patients into overweight, normal, and underweight groups. This analysis did not result in any significant difference between groups. In this sense, we chose not to include it in the article.
To demonstrate this argument, we attach the table produced for this purpose.

Reviewer 4 Report
Comments and Suggestions for Authors
I thank the authors for the review process done.
Author Response
Comments and Suggestions for Authors
I thank the authors for the review process done.
# Authors' Answers
The authors would like to thank the reviewer again for his comments.